# Effect of Atmospheric Pressure Plasma Treatment on Adhesive Bonding of Carbon Fiber Reinforced Polymer

**DOI:** 10.3390/polym11010139

**Published:** 2019-01-15

**Authors:** Chengcheng Sun, Junying Min, Jianping Lin, Hailang Wan

**Affiliations:** 1School of Mechanical Engineering, Tongji University, Shanghai 201804, China; 15902132360@163.com (C.S.); hailang_wan@tongji.edu.cn (H.W.); 2Key Lab of Vehicle Aerodynamics and Vehicle Thermal Management Systems of Shanghai, Tongji University, Shanghai 201804, China

**Keywords:** carbon fiber reinforced polymer, atmospheric pressure plasma treatment, surface modification, shear bond strength

## Abstract

To improve the strength of the adhesive-bonded carbon fiber reinforced polymer (CFRP) joints, atmospheric pressure plasma treatment (APPT) was used to treat a CFRP substrate surface. This study investigated the effects of nozzle distance (i.e., the distance between plasma nozzle and CFRP substrate) and nozzle speed (i.e., the moving speed of plasma nozzle relative to CFRP substrate) of APPT on the lap-shear strength of adhesive-bonded CFRP joints. Results show that the lap-shear strength of plasma-treated CFRP joints increased to a peak value and then decreased as the nozzle distance increased, and the nozzle distance associated with the peaked joint strength depends on the applied nozzle speed. The lap-shear strength of plasma-treated adhesive-bonded CFRP joints reaches up to 31.6 MPa, compared to 8.6 MPa of the as-received adhesive-bonded CFRP joints. The surface morphology of plasma-treated CFRP substrates was investigated by scanning electron microscope observation, and the mechanism associated with the improved joint strength after applying APPT was revealed through surface chemistry analysis. It is found that APPT not only effectively removed the content of Si element and –CH_3_ (i.e., the main compositions of release agent) from the as-received CFRP substrate surface, but also generated many polar groups (i.e., –NH_2_, –OH, –COOH, etc.), which has a positive effect on increasing the wettability and interfacial bonding strength of CFRP substrates and consequently results in a significant improvement of lap-shear strength of plasma-treated CFRP joints. In addition, the result of differential scanning calorimetry (DSC) test shows that the surface temperature of CFRP substrate should not exceed 175.3 °C during APPT. In this study, an empirical model governing temperature, nozzle distance and nozzle speed was established to guide the selection of atmospheric pressure plasma treatment process parameters in industrial manufacture.

## 1. Introduction

Due to the high specific strength, corrosion resistance, and good formability, carbon fiber reinforced polymers (CFRP) are increasingly used, especially in automotive areas and aerospace fields [1,2,3]. Compared with other traditional joining techniques (e.g., riveting, welding, etc.), adhesive bonding does not damage and weaken CFRP substrates. Besides, the adhesive bonding structure not only has the merit of excellent mechanical properties, for example, high shear strength and high fatigue resistance, but also has significant cost advantages. Therefore, it has become a popular method in CFRP joining [4,5]. However, the bonding strength of the as-received CFRP joint is low due to the presence of the mold release agent and other contaminations on the as-received CFRP substrate surface [6]. In order to achieve a high strength adhesive bond, the surface of CFRP substrate needs to be cleaned and activated before applying the adhesive.

The surface treatments of CFRP can be classified into mechanical surface treatments (e.g., manual abrading, grit blasting, etc.) and chemical surface treatments (e.g., acid oxidation, coupling agent treatment, etc.) [7,8,9]. Manual abrading and grit blasting have the disadvantages of damaging carbon fibers and producing contaminations during these processes, which need to be secondly cleaned [10]. The chemical surface treatments have drawbacks such as low efficiency and environmental pollution. Compared with mechanical and chemical surface treatment, plasma surface treatment has attracted more and more attention, because of the advantages of high efficiency and environmental friendliness. Plasma treatment used to have a requirement of low pressure, which resulted in additional costs to ensure the vacuum of the plasma treatment system [11,12]. In addition, the size of the vacuum chamber limited the size and number of pretreated CFRP. With the development of plasma source using at atmospheric environment, APPT overcomes the disadvantage of low pressure plasma treatment and enables the application of this technology in industrial production [13]. Therefore, APPT has become the focus of studying in the adhesive bonding of CFRP.

Several studies [14,15,16,17,18,19] have indicated that the effective application of plasma treatment for the surface modification of polymer materials can improve adhesive bonding strength. Kim et al. [14,15] applied Ar plasma treatment to CFRP surface in a vacuum for studying the influence of process parameters (e.g., voltage, time, and pressure) on the wettability and bonding strength. Their experiment results showed that when the vacuum pressure varied between 400–800 mTorr, power supply varied between 10–30 W, and the treatment time varied between 10–120 s, the plasma surface treatment achieved desired results. Williams et al. [16] used atmospheric pressure helium and oxygen plasma treatment to improve the adhesion between the epoxy composites and stainless steel. Their research showed that after plasma activation, the lap-shear strength had an increase of 80% for stainless steel coupons and 150% for carbon-fiber epoxy laminates. Furthermore, APPT had been used by Zaldivar et al. [17] to treat carbon fiber reinforced epoxy composite, which increased the wettability and carboxyl groups’ concentration of substrate surface, and consequently, the bonding performance of the adhesive-bonded joints increased from 16.5 MPa to 25 MPa. Similar methods were used by Jan et al. [18] to improve the bonding strength of the polyamide 6 composites. Comyn et al. [19] found that plasma treatment could generate rich –CO–, –OH bonds on the substrate surface, which has strong binding force with epoxy adhesive. In addition, plasma-treated CFRP surfaces has good stability, which can be stored under laboratory conditions up to 3 months without obvious loss of enhanced adhesion properties. In our previous study [20], it was found that the nozzle distance (i.e., distance between plasma nozzle and CFRP substrate) had an obvious impact on surface modification of a sheet molding compound (SMC), including remove of release agent, increase in the O and N contents and enhanced wettability. To increase surface treatment efficiency and obtain desired bonding strength of CFRP, it is necessary to study the effect of nozzle distance and nozzle speed (i.e., the moving speed of plasma nozzle relative to CFRP substrate) on the bonding performance of adhesive-bonded CFRP joints. In addition, the existence form of chemical elements on the CFRP surface requires further analysis and the effect of temperature on the bonding strength of CFRP should be well understood.

The objective of this study is to understand the effect of plasma surface treatment on the performance of adhesive-bonded CFRP joints for achieving the industrial automation application of APPT. This study mainly consists of four parts: the first part introduces experimental materials and procedures. The second part describes the experiment results about the lap-shear strength and fracture modes of the adhesive-bonded CFRP joints. The third part analyzes the effect of plasma treatment on CFRP substrate from the aspects of surface morphology, the change of surface chemical functional groups and surface wettability. In the last part, the improving mechanism of APPT on the lap shear strength of adhesive bonded CFRP joints is discussed.

## 2. Experimental Details

### 2.1. Materials

The CFRP used in this study was consisted of E44/51 epoxy resin and T300-12k pan-based carbon fibers. The material properties of the epoxy resin and carbon fibers provided from the suppliers are listed in Table 1 and Table 2, respectively. Carbon fiber prepregs were placed on a die coated with release agent (i.e., dimethyl-silicon oil) and then put in an autoclave to cure. The curing condition was 120 °C for 2 h with a pressure of 0.69 MPa (100 psi). The thickness of CFRP sheet was 1.5 mm, which was consisted of 11 layers of prepreg with laying direction of [0,90]. The CFRP sheet was cut by water jet scalpel into 100 × 25 mm^2^ substrates, where the length direction was consistent with the direction of the most top layer carbon fiber. The adhesive used in this study was two-component 3M DP460, which is a room-temperature-curing epoxy adhesive. The mechanical properties of cured 3M DP460 are listed in Table 3.

### 2.2. Plasma Treatment

The operating frequency and voltage of the APPT equipment are 12 kHz and 8 kV respectively, and the waveform of the operating voltage is a sine wave. From Figure 1, it can be seen that plasma gas is jetted to the surface of CFRP substrate through a circular nozzle with a diameter of 4 mm. Air was used as plasma gas in this study due to easy access, and the gas flow was set to 30 L/min. The motion trail of plasma nozzle relative to CFRP substrate is presented in Figure 2, where CFRP substrates were placed on an electronically-controlled platform. An area of 14 × 25 mm^2^, slightly larger than the bond area (12.5 × 25 mm^2^), was treated by APPT on CFRP substrates. In this study, the distance between plasma nozzle and CFRP substrate (nozzle distance, h) was set to 10 mm, 14 mm, 18 mm, 22 mm and 26 mm, and the moving speed of plasma nozzle relative to CFRP substrate (nozzle speed, v) was set to 1 mm/s, 3 mm/s, 5 mm/s, 7 mm/s and 10 mm/s. The plasma treated CFRP joints were designated as $P_{v-h}$ CFRP joints when the nozzle speed of v and nozzle distance of h were applied to the APPT.

### 2.3. Joint Fabrication

The size of lap-shear adhesive-bonded joint sample is shown in Figure 3 and the fabrication processes are as follows: (1) Cleaning the bonding areas of two CFRP substrates with acetone; (2) Applying the adhesive onto the bonding area of either of the substrates by using an injection gun; (3) Scattering a layer of glass balls (0.25 mm in diameter) on the adhesive surface to control the thickness of the adhesive layer; (The glass balls remain in the final adhesive-bonded joint.) (4) Applying pressure via the fixture and removing the excessive adhesive outside the bond area; (5) Examining joint quality to ensure the consistency of tensile testing results, and then curing the joints in an oven at 40 °C for 2 h per supplier’s recommendation.

### 2.4. Quasi-Static Lap-Shear Tensile Test

The quasi-static lap-shear tensile tests were carried out to measure joint strength according to the standard ASTM D1002-2001 [21]. The MTS E45.105 tensile tester was used to load CFRP adhesive-bonded joints to failure with a crosshead speed of 10 mm/min. Two filler plates were attached to both ends of the adhesive-bonded joint samples to minimize the effects of bending stress generated during the tensile test. The maximum load obtained from tensile tests was divided by the bond area to calculate the lap-shear strength of CFRP joints. The bond area of each joint was subject to the actual measurement and the maximum load was the average of five replicates.

### 2.5. Temperature Measurements

A thermal infrared imager (Optris-PI400) was used to measure CFRP substrate surface temperature via thermal radiation during APPT as shown in Figure 4. It can be connected to a computer via a USB connector and operated in Optris PI Connect software. The applied sampling frequency of the thermal infrared imager is 80 Hz and temperature measurement error is 2 °C. The material emissivity of CFRP is 0.85. In this study, the average temperature of the area directly under the nozzle (i.e., a circle with a diameter of 4 mm) was used as the surface temperature of CFRP substrate during the APPT.

### 2.6. Scanning Electron Microscope (SEM) Observation

The surface morphologies of CFRP substrates were observed with a scanning electron microscope (SEM, Zeiss, Jena, Thuringia, Germany) at a voltage of 15 kV. As-received and plasma-treated CFRP substrates were cut into square specimens of 10 × 10 mm^2^. A thin layer of gold was coated onto the specimen surfaces prior to SEM observation.

### 2.7. X-ray Photo Spectroscopy (XPS) Tests

The surface chemical composition of CFRP substrates was examined using a Thermo Fisher Scientific’s ESCALAB 250 series X-ray photo electron spectroscopy (XPS, Thermo Fisher Scientific, Waltham, MA, USA). The apparatus used an Al Kα monochromatic X-ray source and its analysis area was 900 μm^2^. The base pressure and the energy resolution of the XPS test system were set to 4.3 × 10^−8^ Pa and 0.44 eV, respectively.

### 2.8. Fourier Transform Infrared Spectroscopy (FTIR) Tests

The evolution of functional groups on the surface of CFRP substrates caused by APPT was analyzed using a Nicolet’s AVATAR 360 Fourier transform infrared spectroscopy (FTIR, Thermo Nicolet Corporation, Madison, WI, USA). Each sample was scanned for 32 times and the resolution was 4 cm^−1^. The infrared spectrum ranged from 400 cm^−1^ to 4000 cm^−1^.

### 2.9. Contact Angle Measurements

To investigate the effect of APPT on the surface wetting properties of CFRP substrates, Dataphysics OCA-20 contact-angle analyzer was used to measure contact angles of CFRP substrate surfaces. Three typical test liquids (i.e., Deionized water, diiodomethane and ethylene glycol) with known dispersion component and polar component [22], were selected to calculate the surface free energy of CFRP substrates. The volume of test liquid was controlled as 2 μL. The contact angles were read when the test liquid stabilized on the surface of CFRP substrate, and their values were averaged from three replicates.

### 2.10. Differential Scanning Calorimetry (DSC) Tests

Differential scanning calorimetry (DSC, TA Instruments, Newcastle, DE, USA) was tested on a TA Instruments DSC Q100 to investigate the chemical reaction temperature of epoxy resin. Epoxy resin with a weight of 4.810 mg was placed in a hermetic crucible, and an empty crucible was used as a reference. The temperature range of quantitative dynamic scanning was from 0 °C to 250 °C at a heating rate of 10 °C/min.

### 2.11. Thermogravimetric Analysis (TGA)

Thermogravimetric Analysis (TGA, TA Instruments, Newcastle, DE, USA) was used to investigate the weight change of CFRP as the temperature increases. CFRP substrate with a weight of 8.120 mg was placed in a crucible and then put in a heating unit. The temperature range in the heating unit was from 20 °C to 600 °C and the heating rate was 10 °C/min.

## 3. Results

### 3.1. Lap-Shear Tensile Testing Results

To evaluate the effect of APPT on the bonding strength, adhesive bonded CFRP joints were fabricated and lap-shear tensile tests were conducted. Figure 5 presents the lap-shear strength of adhesive-bonded CFRP joints as a function of v and h. It can be seen that the lap-shear strength of adhesive-bonded joints fabricated from as-received CFRP substrates was only 8.6 MPa. When v was 1 mm/s, the lap-shear strength of adhesive-bonded joints fabricated from plasma-treated CFRP substrates ($P_{1-h}$ CFRP joints) increased from 21.2 MPa to 30.5 MPa as h increased from 10 mm to 18 mm. While h was beyond 18 mm, the lap-shear strength of plasma-treated CFRP joints decreased nearly linearly with an increase of h. Similar effect of h on the lap-shear strength of $P_{v-h}$ CFRP joints was observed as v was set to 5 mm/s, at which the lap-shear strength of $P_{5-h}$ CFRP joints reached a peak value of 31.6 MPa when h was 18 mm. When v was 10 mm/s, the lap-shear strength of $P_{10-h}$ CFRP joints increased from 27.7 MPa to 29.7 MPa as h increased from 10 mm to 14 mm, then decreased continuously with a further increase of h. Compared with the as-received CFRP joints, the lap-shear strength of P_1-18_ and P_10-14_ CFRP joints had an increase of 255% and 245%, respectively, which are slightly lower than that of P_5-18_ CFRP joints (267%).

In addition to the lap-shear strength, the fracture modes of adhesive-bonded joints are always regarded as an important aspect to evaluate their bonding performance. The fracture modes of plasma-treated CFRP joints can be classified into three scenarios: substrate failure (failure of CRFP substrate), cohesive failure (failure occurring inside the adhesive), and adhesive failure (failure at the interface between adhesive and substrate). Table 4 shows a schematic diagram of the three fracture modes and Table 5 listed the fracture modes of all plasma-treated CFRP joints after lap-shear tensile testing. From Table 5, it can been seen that the fracture modes of P_1-10_, P_1-14_, P_1-18_, P_5-10_, P_5-14_, P_10-10_, and P_10-14_ CFRP joints were substrate failure, where a layer of broken carbon fibers was stuck on the adhesive. Therefore, the lap-shear strength of joints was dominated by the strength of the CFRP substrate. The fracture mode of P_5-18_ CFRP joint was cohesive failure. For the cohesive failure, the lap-shear strength of adhesive bonded joints was mainly dependent on the shear strength of adhesive, which indicates that it is not necessary to continue to activate the CFRP substrate surface since the interfacial bonding strength between adhesive and CFRP substrate is higher than the shear strength of adhesive. The fracture modes of P_1-22_, P_1-26_, P_5-22_, P_5-26_, P_10-18_, P_10-22_, and P_10-26_ CFRP joints were adhesive failure similar to that of the as-received CFRP joints. In this case, the lap-shear strength of adhesive bonded joints was dominated by the interfacial bonding strength between the adhesive and CFRP substrate. The lap-shear strength of adhesive bonded joints in the case of cohesive failure was higher than the failure of the CFRP substrate, which may be due to thermal damage (i.e., pyrolysis carbonization of resin matrix) introduced to the CFRP substrate caused by APPT as smaller h and lower v were applied [23,24].

### 3.2. Temperature Measurement Results

In order to understand how the plasma processing parameters affect the surface temperature of CFRP substrate (T), T was measured by a thermal infrared imager when the CFRP substrate was treated by plasma equipment with different v and h, and the measured temperatures $\left(T_{\exp}\right)$ are presented in Figure 6. From Figure 6, it can be seen that the highest $T_{\exp}$ occurred at the smallest h and the lowest v. It is found that h showed a more significant effect on the $T_{\exp}$ than v. When v was 1 mm/s, $T_{\exp}$ decreased from 538 °C to 132 °C as the h increased from 10 mm to 26 mm. When h was 10 mm, the $T_{\exp}$ decreased from 538 °C to 437 °C as the v increased from 1 mm/s to 10 mm/s. In addition, the effect of v on the $T_{\exp}$ decreased with increasing h. When h was 26 mm, the $T_{\exp}$ decreased by only 30 °C as the v increased from 1 mm/s to 10 mm/s. It is found that T can be expressed as an function of h and v, i.e., Equation (1), and the goodness of fit for T is characterized by the mean square root (MSR) error and maximum deviation (MD). Based on Equations (2) and (3), the MSR error and MD between the calculated temperature by Equation (1) $T_{\mathrm{cal}}$ and $T_{\exp}$ is 5.3 °C and 9.0 °C, respectively. Figure 7 shows that the empirical model (Equation (1)) is capable of providing a satisfied prediction to the surface temperature of CFRP substrate at least within the investigated ranges of v and h.
(1)$$T=\left(-5.90*10^{-3}*h^{3}+3.24*10^{-1}*h^{2}-5.05*h+12.71\right)*v-1.54*\;10^{-1}*h^{3}\;+1.04*10^{1}*h^{2}-2.40*10^{2}*h+2.06*10^{3}$$
(2)$$\mathrm{MSR}=\sqrt{\frac{1}{25}*\overset{n=25}{\underset{n=1}{\sum }}{\left(T_{\mathrm{cal}}-T_{\exp}\right)}^{2}}$$
(3)$$\mathrm{MD}=\mathrm{Max}\overset{n=25}{\underset{n=1}{\sum }}|T_{\mathrm{cal}}-T_{\exp}|$$

### 3.3. Surface Morphologies of CFRP Substrates

To examine the morphological changes in the surface of plasma-treated CFRP substrate, scanning electron microscopy (SEM) observations were performed. Figure 8a–f present the surface morphologies of the as-received and plasma-treated CFRP substrates (i.e., P_5-10_, P_5-14_, P_5-18_, P_10-14_, and P_10-18_ CFRP substrates). From Figure 8a, it can be found that the surface of as-received CFRP substrate is relatively homogeneous and smooth, but there are some tiny pores having a diameter of smaller than 2 μm on the surface of the epoxy resin. When v of 5 mm/s and h of 10 mm were applied to treating CFRP substrate, it can be observed from Figure 8b that the epoxy resin on the surface of P_5-10_ CFRP substrate (with $T_{\exp}$ = 482 °C) was removed and carbon fibers on the outermost layer were exposed and had minor damage. A lot of pores larger than 5 μm in diameter on the epoxy resin of P_5-14_ (with $T_{\exp}$ = 246 °C) and P_10-14_ (with $T_{\exp}$ = 208 °C) CFRP substrates were observed, as shown in Figure 8c,d, respectively. Some scholars have reported that double bonds can be formed by eliminating water in epoxy resin molecules at the temperature higher than 120 °C [25], which may create larger and deeper pores on the original pores or create new pores in the surface of the epoxy resin. The surface morphologies of P_5-18_ (with $T_{\exp}$ = 174 °C) and P_10-18_ (with $T_{\exp}$ = 138 °C) CFRP substrates were nearly the same as that of the as-received CFRP substrate shown in Figure 8e,f. Therefore, it is suggested that there is little thermal damage occurred on the CFRP substrates when the surface temperature of CFRP substrate during APPT is below 174 °C.

### 3.4. Surface Chemistries of CFRP Substrates

To study the chemical element changes caused by APPT on CFRP substrates, X-ray photoelectron spectroscopy (XPS) was used to detect the surface chemical compositions of the as-received and plasma-treated CFRP substrates (i.e., P_5-18_, P_10-18_, and P_5-26_ CFRP substrates). Figure 9 presents the high resolution XPS scanning results of Si2s, Si2p, C1s, N1s, and O1s. The content of each element of interest is listed in Table 6. It can be seen from the table that compared with the as-received substrate, the content of Si element on the P_5-18,_ P_10-18_ and P_5-26_ CFRP substrate surfaces was reduced by 70%, 51%, and 15% after APPT, respectively. The Si element, mainly existing in the form of –(CH_3_)_2_Si–O–, is one of the components of the release agent, which acts as surface contamination on the CFRP substrate blocking the bonding of the adhesive and epoxy resin [26]. From the reduced Si contents on the plasma-treated CFRP substrate surfaces, the contaminations on the CFRP substrate surface can be removed more completely by APPT when a smaller h or a slower v was applied. Comparing the contents of O and N elements on the P_5-18_ and P_5-26_ CFRP substrate surfaces, it can be found that when h was increased from 18 mm to 26 mm, the contents of oxygen and nitrogen dropped by 56% and 45%. While h was fixed at 18 mm and v was increased from 5 mm/s to 10 mm/s, it is observed that the contents of oxygen and nitrogen dropped by 21% and 39%. This suggests that the increase of either h or v results in reducing the contents of both oxygen and nitrogen on the plasma-treated CFRP substrate surface.

Figure 10 shows the carbon functional groups XPS analyses of the as-received and plasma-treated (i.e., P_5-18_ CFRP substrates, the chemical element change on the surface is most obvious) CFRP substrates surface. The peaks at 288.7 eV and 286.2 eV are attributed to C=O (or O–C=O) and C–O (or C–N), respectively, and the peak at 284.7 eV is representative of C–C (or C–H). Table 7 shows the content of carbon functional groups. It can be observed that after APPT with v = 5 mm/s and h = 18 mm, the C=O/O–C=O bond content of CFRP substrate surface was increased by 118.4%, the C–O/C-N bond content of CFRP substrate surface was increased by 139.5%, while the C–C/C–H bond content of CFRP substrate surface was decreased by 31.7%.

FTIR was performed to further analyze surface modifications of plasma-treated CFRP substrates as well as to identify what functional groups contribute to improve interface bond strength. Except for the observed differences within the range from 2600 to 3800 cm^−1^, the overall absorbance of P_5-18_ CFRP substrate surface was almost the same with the as-received substrate surface. Figure 11 presents a compilation of the spectra as an effect of APPT within this range. The absorption between 2800 and 3000 cm^−1^ is related to –CH_3_, which is typically quite stable on the surface of the as-received CFRP substrate even after curing process of CFRP sheet or acetone-cleaning. However, after APPT –CH_3_ groups appeared to become oxidized and the intensity decreased. In addition, the absorptions at 3500 cm^−1^ and 3300 cm^−1^ are associated with –OH and –NH_2_, and both had a significant increase in intensity.

### 3.5. Wettability of CFRP Substrates

To examine the effect of APPT on wetting characteristics of CFRP substrates, the contact angles of as-received and P_5-18_ CFRP substrates were compared. Before measuring contact angles, as-received CFRP substrates were cleaned with acetone and dried in laboratory environment. The wettability of CFRP substrates can be evaluated through the measured contact angles of the substrate surfaces [27]. As listed in Table 8, APPT decreased the contact angles for all test liquids (i.e., distilled water, ethylene glycol, and diiodomethane). When distilled water was used as the test liquid, the contact angles of as-received and P_5-18_ CFRP substrates were presented in Figure 12. The contact angle of the as-received CFRP substrate was measured as 108.6°, while the contact angle decreases to 32.4° after APPT with v = 5 mm/s and h = 18 mm (i.e., the P_5-18_ CFRP substrate). These results indicate that the surfaces of CFRP substrates became more hydrophilic after APPT. Similar experimental phenomena were observed in previous reports [28,29].

To further evaluate surface wetting properties, the surface free energy (SFE) of CFRP substrates, expressed by Equation (4), was calculated based on the Owens–Wendt method [30,31]. Figure 13 presents the surface free energy consisted of corresponding polar and dispersion components. The SFE of the P_5-18_ CFRP substrate increased significantly compared to the as-received substrate, which agrees well with the reported results based on other plasma pretreatment methods [32]. The SFE of the P_5-18_ CFRP substrate reached up to 61.33 mJ/m^2^ and was enhanced by 36.81 mJ/m^2^ compared to that of the as-received CFRP substrate. Referring to Figure 13, the enhancement of SFE of the P_5-18_ CFRP substrate is mainly due to the increase of its polar component.
(4)$$\gamma_{L}\left(1+\cos\alpha \right)=2\sqrt{\gamma_{S}^{d}\gamma_{L}^{d}}+2\sqrt{\gamma_{S}^{p}\gamma_{L}^{p}}$$

In the Equation (4), $\gamma_{L}$ represents the surface energy of test liquids, $\gamma_{L}^{d}$ and $\gamma_{L}^{p}$ are dispersion and polar components of $\gamma_{L}$, respectively; $\gamma_{S}$ represents the surface energy of CFRP substrate, $\gamma_{S}^{d}$ and $\gamma_{S}^{p}$ are dispersion and polar components of $\gamma_{S}$, respectively; and α represents the measured contact angle.

### 3.6. Thermal Damage Temperature of CFRP Substrates

To study the effect of temperature on the CFRP substrate, the epoxy resin of CFRP was tested by DSC. The DSC testing result is presented in Figure 14. From Figure 14, it can be seen that the cured epoxy resin has an endothermic peak at 194.5 °C and the initial temperature of the endothermic peak is 175.3 °C. This indicates that the epoxy resin started to react chemically as the temperature reached 175.3 °C. Figure 15 presents the weight of CFRP substrate (m) changes as a function of temperature. The weight of the CFRP substrate started to decrease as the temperature was above *T*_1_ (~170 °C). When the temperature reached *T*_2_ (~340 °C), the epoxy resin in the CFRP substrate was almost fully pyrolysed.

## 4. Discussion

From the above results, the mechanism accounting for the improved lap-shear strength of CFRP adhesive bonded joints through APPT is discussed here. On the one hand, APPT has a cleaning effect of surface contaminations on the CFRP substrates. Compared to the as-received CFRP substrate, the content of Si element on the plasma-treated CFRP substrate surface was decreased significantly. In addition, the –CH_3_ groups, which are unfavorable for adhesive bonding as non-polar groups [33], were oxidized or removed after APPT. These indicate that there was a layer of release agent remaining on the surface of the as-received CFRP, of which the main component is –(CH_3_)_2_Si–O– that cannot be removed by acetone cleaning. Release agent is an inert substance blocking the bond between the adhesive and CFRP substrate, which has been reported resulting in low bonding strength and adhesive failure of as-received CFRP joint [26]. Therefore, the lap-shear strength of adhesive bonded joints increased after removing the release agent by APPT. On the other hand, APPT generates many polar groups such as –OH, –COOH, and –NH_2_ on the surface of CFRP substrates. Compared to the as-received CFRP substrate, APPT increased the contents of O and N elements on the plasma-treated CFRP substrate surface. Further analysis revealed that the O element mainly exists in the form of –OH and –COOH, while the N element mainly exists in the form of –NH_2_. These polar groups may form hydrogen bonds and covalent bonds with the adhesive molecules [34,35] as illustrated in Figure 16, which therefore improves the lap-shear strength of the adhesive interface. In addition, it can be concluded from Figure 11 that the surface of plasma-treated CFRP substrate is more hydrophilic, which provides a better bonding condition between the adhesive and CFRP substrate surface and consequently further improves the lap-shear strength of the adhesive-bonded joints. Figure 17 summarizes the above discussion in order to better understand the effect of APPT on the CFRP substrate surface.

It can be seen from the changes in the contents of Si, –CH_3_, –OH, –COOH, and –NH_2_ after APPT, the cleaning effect of surface release agent and surface activation with increasing polar groups would be improved as v and/or h decreased, which consequently contributed to the increase in lap-shear strength of plasma-treated CFRP joints. However, the lap-shear strength of plasma-treated CFRP joints would decrease when a slow v (e.g., v < 5 mm/s) and/or a small h (e.g., h < 18 mm) were applied, which was mainly attributed to the thermal damage occurring on CFRP substrates caused by high temperatures. The thermal damage decreases the mechanical properties of CFRP substrate and results in the failure of CFRP substrate during lap-shear tensile testing, which consequently decreases the lap-shear strength of plasma-treated CFRP joints. The DSC testing result suggests that the epoxy resin began to react chemically when the temperature was over 175.3 °C and TGA result shows that the epoxy resin was almost completely pyrolysed when the temperature reached about 340 °C. Therefore, there were many pores on the surface of the epoxy resin (e.g., in Figure 8d) when the surface temperature of the CFRP substrate was 208 °C during APPT; while the CFRP substrate surfaces in Figure 8e did not exhibit obvious pores after APPT where the surface temperature was only 174 °C. As the temperature rose further to 482 °C, the epoxy resin was removed and carbon fibers were exposed.

Therefore, to effectively remove surface contaminates, generate sufficient polar groups, and simultaneously avoid introducing thermal damage to the CFRP substrates, the surface temperature of CFRP substrate should be controlled between 138 °C and 175.3 °C during APPT. The relationship among the surface temperature of CFRP substrate, v and h has been established based on the empirical model, i.e., Equation (1), which can be applied to the guidance of selecting the parameters v and h in the APPT of CFRP substrates.

## 5. Conclusions

In this study, the effect of atmospheric pressure plasma treatment on the static lap-shear strength of the adhesive-bonded CFRP joints was investigated. The mechanism was discussed based on various experimental evidences, and conclusions can be drawn as follows:Atmospheric pressure plasma treatment increases the lap-shear strength of adhesive-bonded CFRP joints by 267% (from 8.6 MPa to 31.6 MPa) when a nozzle speed of 5 mm/s and a nozzle distance of 18 mm were applied, which is attributed to both the removal of surface contaminants (e.g., release agent) and the generation of polar groups (e.g., –OH, –COOH, and –NH_2_). Further study is required to quantify the contribution of each aspect to the improved CFRP joint strength.During the plasma treatment process, the surface temperature of CFRP substrate should be controlled between 138 °C and 175.3 °C. When the temperature is below 138 °C, favorable surface modification by plasma treatment is not obvious. A temperature higher than 175.3 °C causes the thermal damage of CFRP substrate.The surface temperature of CFRP substrate during APPT is expressed as a function of nozzle distance and nozzle speed to guide the selection of plasma treatment process parameters.

## Figures and Tables

**Figure 1 polymers-11-00139-f001:**
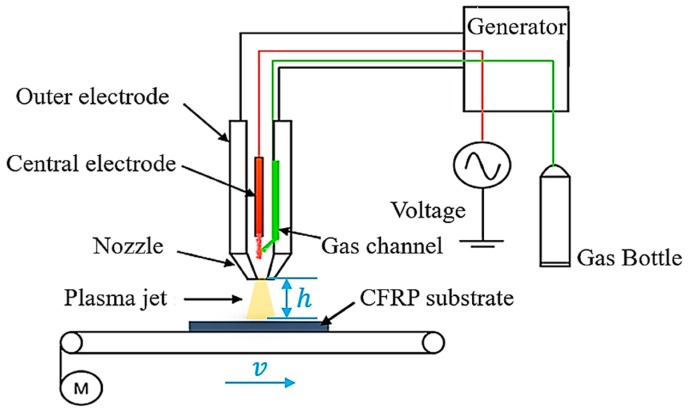
Diagrammatic sketch of the atmospheric pressure plasma treatment equipment.

**Figure 2 polymers-11-00139-f002:**
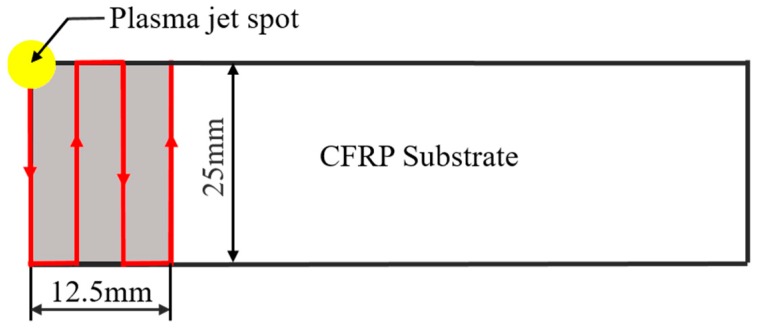
Illustration of the relative motion path between plasma jet and carbon fiber reinforced polymers (CFRP) substrate.

**Figure 3 polymers-11-00139-f003:**
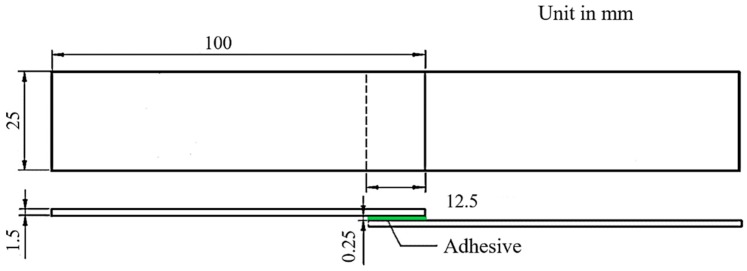
Configuration of the adhesive-bonded CFRP joint.

**Figure 4 polymers-11-00139-f004:**
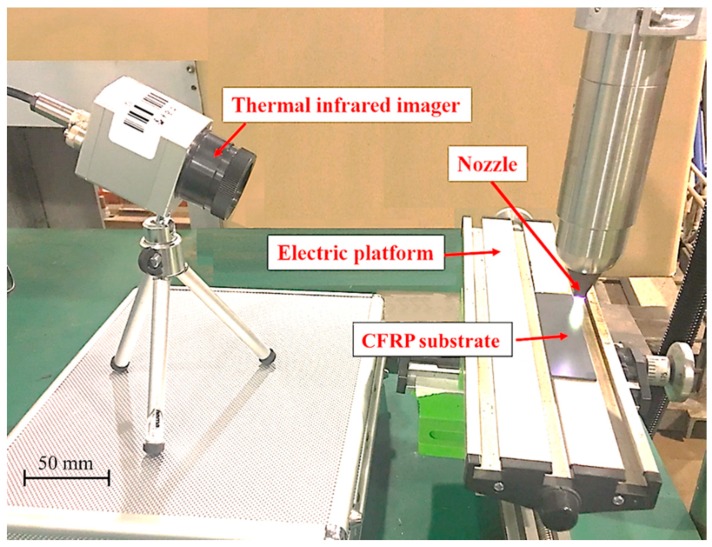
Measuring the surface temperature of a CFRP substrate during atmospheric pressure plasma treatment (APPT).

**Figure 5 polymers-11-00139-f005:**
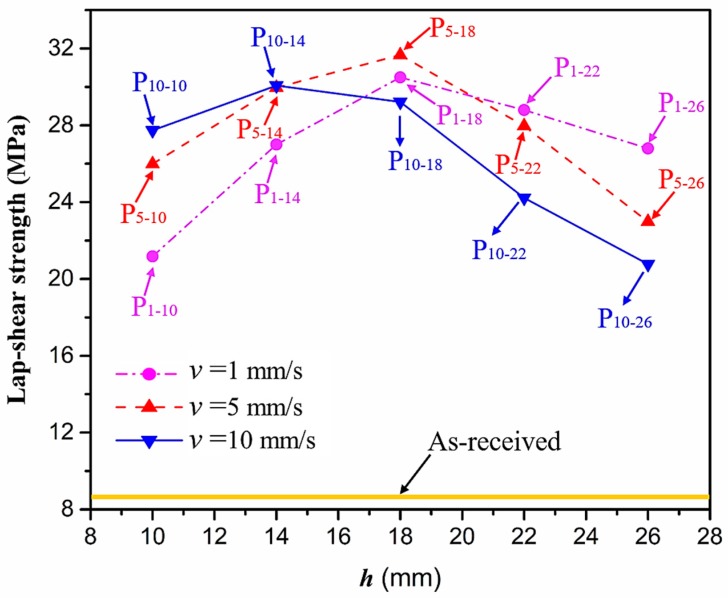
The lap-shear strength of adhesive-bonded CFRP joints as a function of the nozzle speed (v) and the nozzle distance (h).

**Figure 6 polymers-11-00139-f006:**
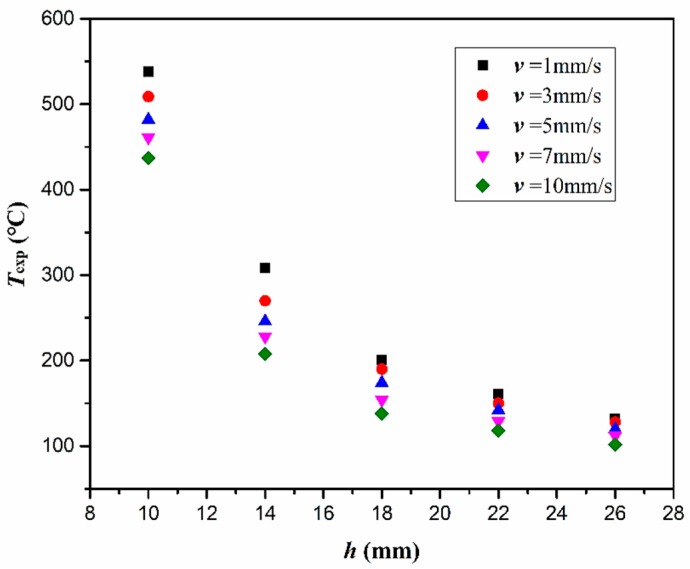
The measured temperature of CFRP substrate surface ($T_{\exp}$) as a function of nozzle speed (v ) and nozzle distance (h).

**Figure 7 polymers-11-00139-f007:**
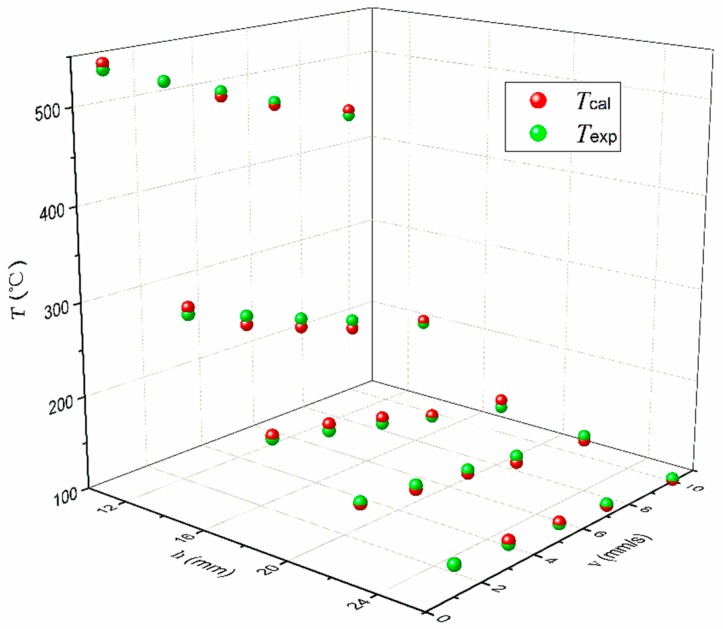
Deviation between the calculated temperature by Equation (1) ($T_{\mathrm{cal}}$) and the experimentally measured temperature ($T_{\exp}$) of CFRP substrates surface.

**Figure 8 polymers-11-00139-f008:**
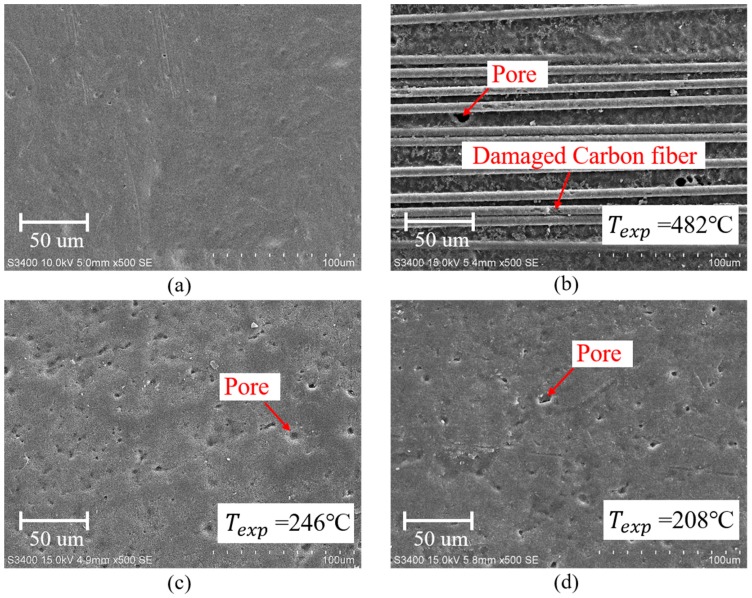
SEM images of (**a**) the as-received CFRP substrate surface and plasma-treated CFRP substrate surfaces of (**b**) P_5__-10_, (**c**) P_5__-14_, (**d**) P_10__-14_, (**e**) P_5__-18_, and (**f**) P_10__-18_.

**Figure 9 polymers-11-00139-f009:**
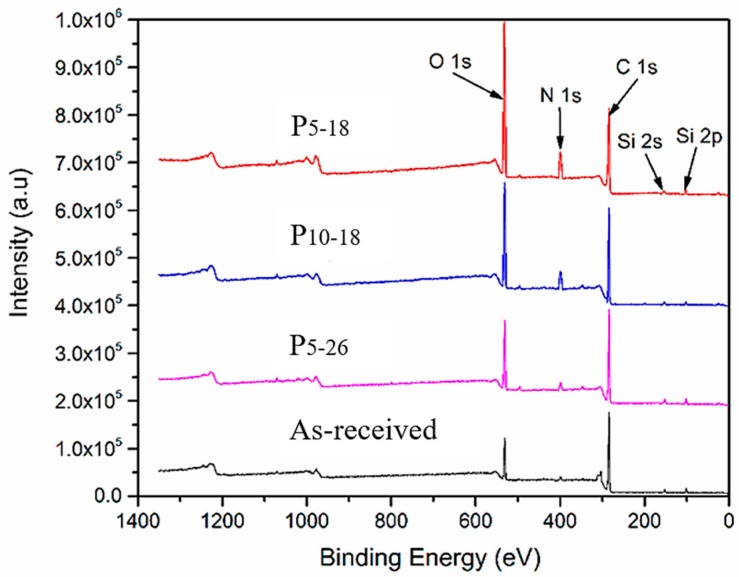
X-ray photoelectron spectroscopy (XPS) spectra showing the effect of plasma treatment process parameters on the surface chemistry of CFRP substrate.

**Figure 10 polymers-11-00139-f010:**
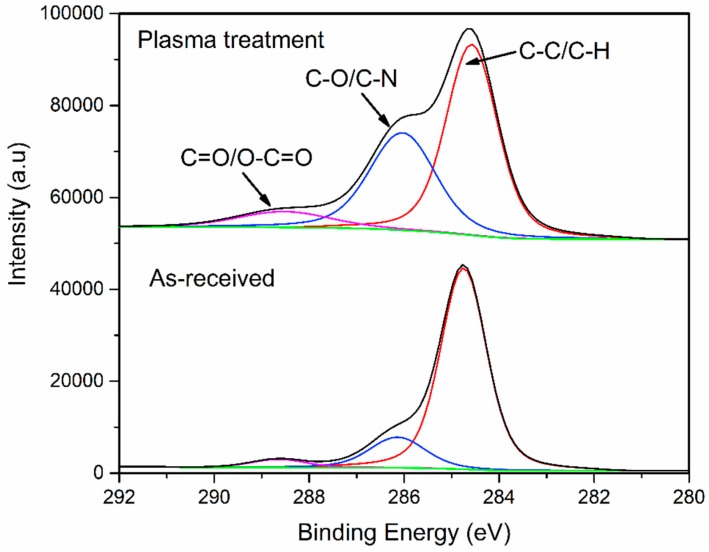
High resolution XPS scans of C 1 s on the as-received and plasma-treated (P_5-18_) CFRP substrate surfaces.

**Figure 11 polymers-11-00139-f011:**
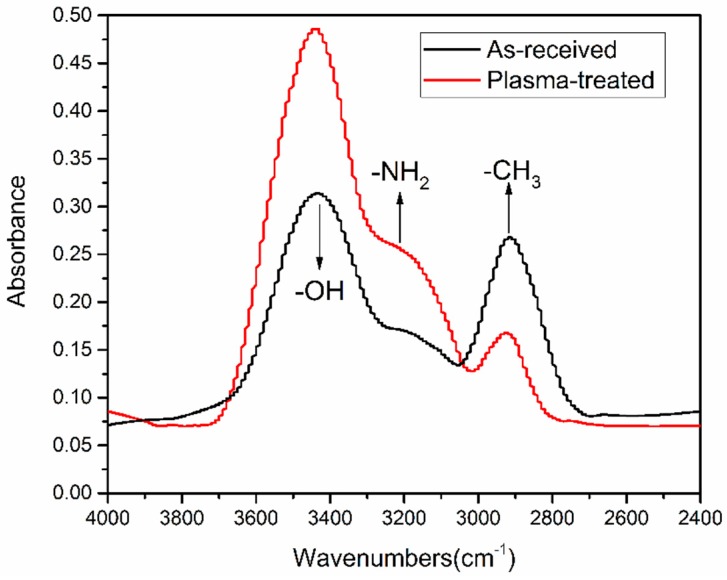
FTIR spectra of the as-received and plasma-treated (P_5-18_) CFRP substrates.

**Figure 12 polymers-11-00139-f012:**
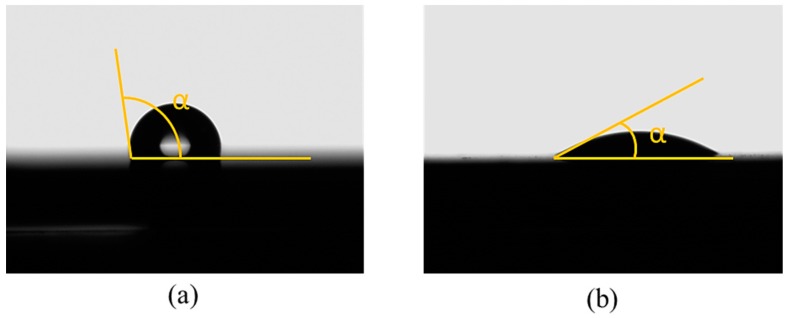
The contact angles of distilled water measured for (**a**) the as-received and (**b**) plasma-treated (P_5-18_) CFRP substrates.

**Figure 13 polymers-11-00139-f013:**
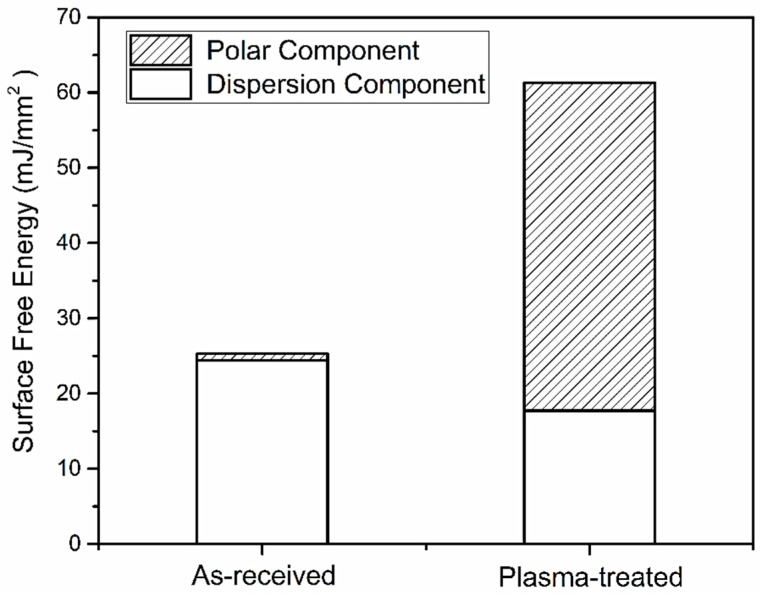
The surface free energies of the as-received and plasma-treated (P_5-18_) CFRP substrates.

**Figure 14 polymers-11-00139-f014:**
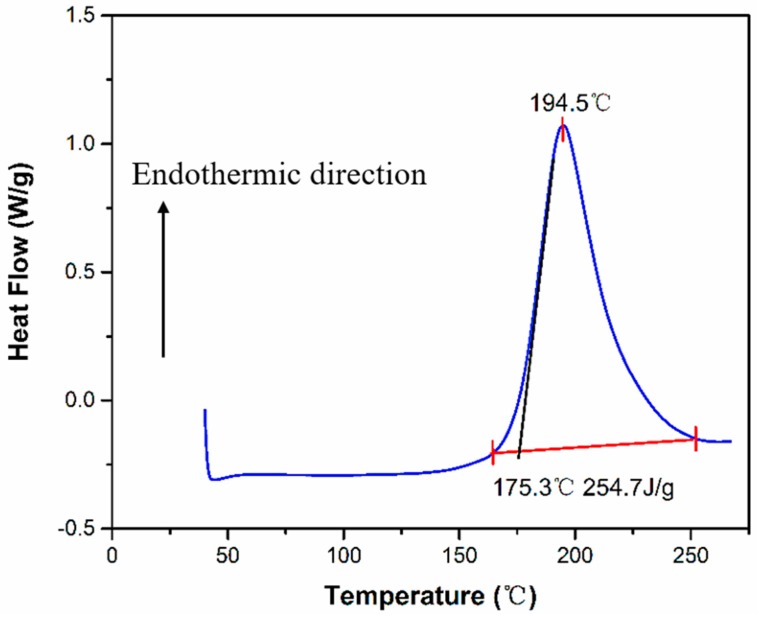
Differential scanning calorimetry (DSC) thermogram of the CFRP epoxy resin.

**Figure 15 polymers-11-00139-f015:**
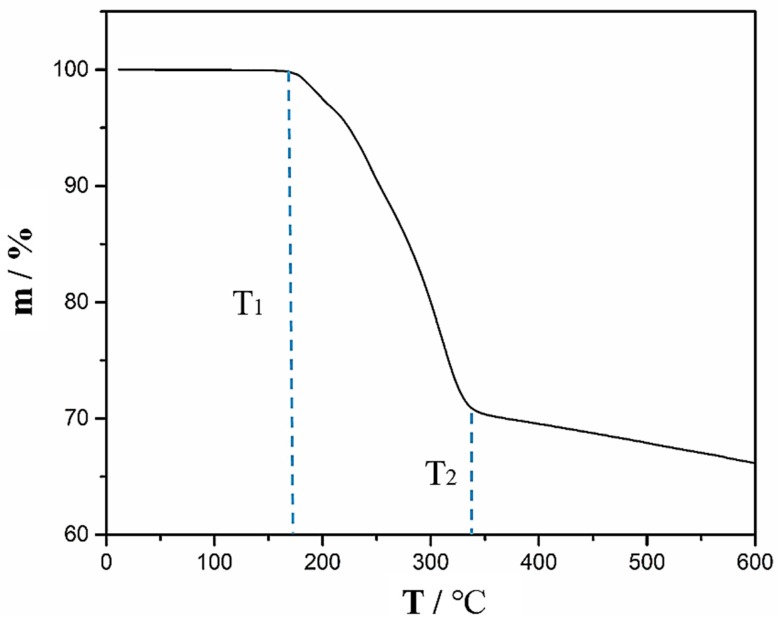
Thermogravimetric Analysis (TGA) showing the change of CFRP substrate weight with increasing temperature.

**Figure 16 polymers-11-00139-f016:**
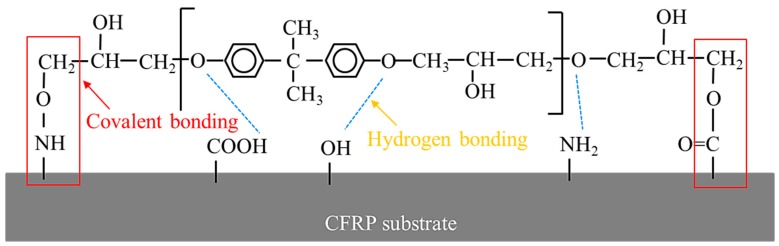
Activated groups and adhesive molecules forming bonding on the CFRP substrate surface.

**Figure 17 polymers-11-00139-f017:**
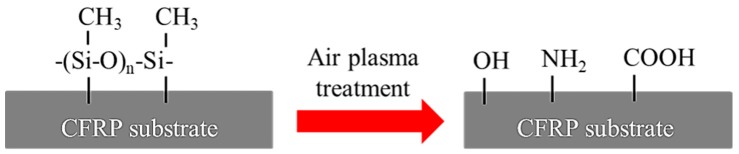
Schematic diagram of atmospheric pressure plasma treatment for the modification of CFRP surface.

**Table 1 polymers-11-00139-t001:** Material properties of carbon fiber.

Material	Fiber Diameter (μm)	Density (g/cm^3^)	Tensile Strength (MPa)	Elongation (%)
Carbon fiber	7	1.76	3530	1.5

**Table 2 polymers-11-00139-t002:** Material properties of epoxy resin.

Material	Density (g/cm^3^)	Specific Heat (J/(kg·°C))	Curing Temperature (°C)	Vaporization Temperature (°C)
Epoxy resin	1.20	1000	120	250

**Table 3 polymers-11-00139-t003:** Mechanical properties of fully cured 3M DP460 adhesive.

Adhesive	Elastic Modulus (GPa)	Tensile Strength (MPa)	Shear Strength (MPa)	Elongation (%)
3M DP460	2.7	37	32	4

**Table 4 polymers-11-00139-t004:** Schematic diagram of fracture modes.

Fracture Modes	Substrate Failure	Cohesive Failure	Adhesive Failure
Schematic diagrams	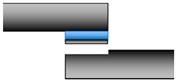	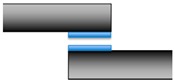	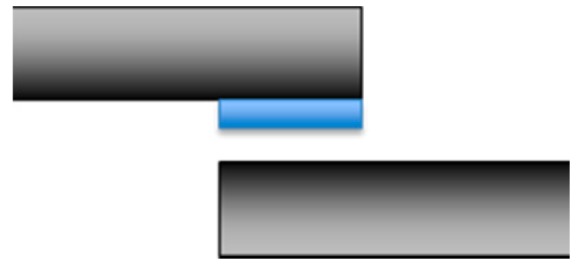

**Table 5 polymers-11-00139-t005:** Fracture modes of plasma-treated CFRP joints.

h (mm)	v (mm/s)		
	1	5	10
10	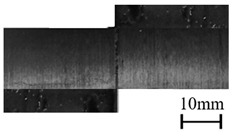 Substrate failure	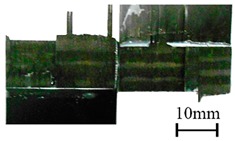 Substrate failure	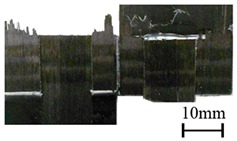 Substrate failure
14	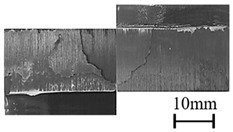 Substrate failure	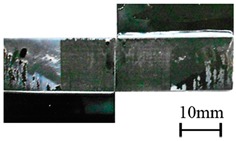 Substrate failure	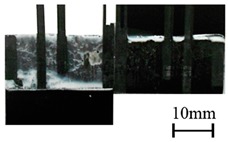 Substrate failure
18	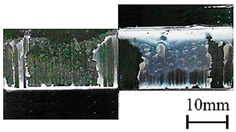 Substrate failure	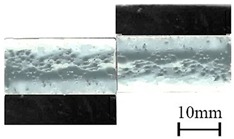 Cohesive failure	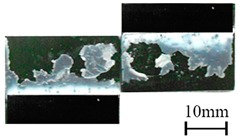 Adhesive failure
22	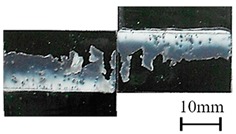 Adhesive failure	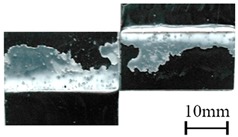 Adhesive failure	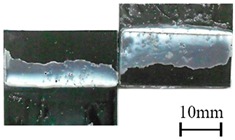 Adhesive failure
26	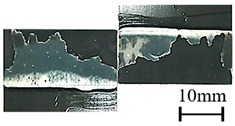 Adhesive failure	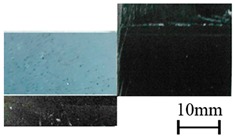 Adhesive failure	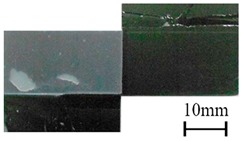 Adhesive failure

**Table 6 polymers-11-00139-t006:** The chemical compositions of CFRP substrate surfaces.

Chemical Element (wt %)		C	O	N	Si
**CFRP substrates**	As-received	74.61	17.69	3.92	3.41
	P_5-18_	63.72	28.88	5.68	1.02
	P_10-18_	67.94	25.12	4.16	1.66
	P_5-26_	72.99	18.93	3.93	2.89

**Table 7 polymers-11-00139-t007:** XPS peak fitting method describing carbon functional group distribution.

Functional Groups (wt %)		C=O/O–C=	C–O/C–N	C–C/C–H
**CFRP substrates**	As-received	3.8	15.2	81.0
	P_5-18_	8.3	36.4	55.3

**Table 8 polymers-11-00139-t008:** Measured contact angles of the as-received and P_5-18_ CFRP substrates.

Testing Liquids		Distilled Water	Diiodomethane	Ethylene Glycol
**CFRP substrates**	As-received	108.6° ± 2.1°	67.1° ± 3.2°	89.7° ± 2.7°
	P_5-18_	32.4° ± 3.3°	54.2° ± 1.6°	43.1° ± 4.1°

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
