# Peer review of "Effect of Atmospheric Pressure Plasma Treatment on Adhesive Bonding of Carbon Fiber Reinforced Polymer"

_polymers, 2019, doi:10.3390/polym11010139_

Round 1

Reviewer 1 Report

The submitted research is interesting at the technical point of view.  I have just few comments as follows.

1. Line 100: What was the wave form of operating voltage?

2. Line 109, Fig. 1: What meaning of Reactor in figure.

3. Line 190 Table 4: These photographs are difficult to understand the structure.  Explanations are needed.

4. Line 225 Table 6: Table 6 and Fig. 6 are the same information.  Teble 6 is not needed.

Author Response

Dear reviewer 1:

         Firstly,we would like to thank you for your suggestive comments. We have carefully revised our manuscript according to the comments. The attachment is my detailed responses to your comments.

Yours,

Junying Min

Reviewer 2 Report

-The introduction states near the end that the influence of two factors (nozzle distance and nozzle speed) needs to be studied. It does not explain why, though. Add a description/explanation to this effect.

-Experimental: Add more details about the materials involved. Later, it is argued that the Si plays an important role. Yet, no Si is mentioned here in the description of the materials.

-Add a reason why air was used as plasma gas.

-Joint fabrication: the addition of glass balls is mentioned. Clarify, whether the glass balls remain within the final bonded joint.

-It would be advisable to first show table 5 and then table 4 so that the reader unfamiliar with the failure modes can understand the description of table 4.

-In table 6, nozzle speeds of 1, 3 and 7 are mentioned. In the experimental part, only speeds of 5 and 10 were described. Amend the experimental part.

-Figure caption, fig. 9: This is not the "Effect of ... parameters" but rather "XPS spectra showing the effect..."

--Figure caption, fig. 10: This is not a survey spectrum.

-There should alway be a space between number and unit.

-The DSC result appears within the discussion section. This should be moved to the results part.

-More diligence should be devoted to the references. For instance, ref. 1: volume and page missing; ref. 2: author missing; ref. 6: wrong year

-You (or your labs) seem to have published a similar paper recently, in Automotive Innovation, July 2018, Volume 1, Issue 3, pp 237–246: Effect of Atmospheric Pressure Plasma Treatment on the Lap-Shear Strength of Adhesive-Bonded Sheet Molding Compound Joints. This paper reports several results which are supposed to be new here, e.g. the lower Si content on the surface after plasma treatment, the higher N and O content, different bonding, reduced contact angle etc. I suggest to include a reference to this paper and to clearly state in this manuscript what is already known, what is new here and also what the significance of the new results is.

Author Response

Dear reviewer 2,

       Firstly, we would like to thank you for your suggestive comments.  We have carefully revised our manuscript according to the comments. The attachment is our detailed responses to the comments.

Yours,

Junying Min

Round 2

Reviewer 2 Report

-Include telephone number in the Correspondence data or delete the "Tel.: +xx-xxx-xxx-xxxx"

-Table 4: caption and content should not be split over two pages.

-Ref. 1: Volume and page still missing. The "S1270963818300774" is not volume/page.

-According to the references guideline, all the authors should be listed, up to ten authors; only then an "et al." can replace the last names